# Spatio-Temporal Changes in Ecosystem Quality across the Belt and Road Region

**DOI:** 10.3390/s23187752

**Published:** 2023-09-08

**Authors:** Xiangqin Wei, Tianhai Cheng, Jian Yang, Shijiao Qiao, Li Li, Haidong Yu, Xiaofei Mi, Yan Liu, Hong Guo, Jiaguo Li, Yuan Sun, Chunmei Wang, Xingfa Gu

**Affiliations:** 1Aerospace Information Research Institute, Chinese Academy of Sciences, Beijing 100094, China; weixq@aircas.ac.cn (X.W.); chength@aircas.ac.cn (T.C.); yangjian@aircas.ac.cn (J.Y.); lilifs@aircas.ac.cn (L.L.); yuhaidong20@mails.ucas.ac.cn (H.Y.); mixf@aircas.ac.cn (X.M.); liuyan@aircas.ac.cn (Y.L.); guohong@aircas.ac.cn (H.G.); lijg@aircas.ac.cn (J.L.); sunyuan@aircas.ac.cn (Y.S.); wangcm@aircas.ac.cn (C.W.); 2State Key Laboratory of Remote Sensing Science, Faculty of Geographical Science, Beijing Normal University, Beijing 100875, China; 202321051206@mail.bnu.edu.cn; 3University of Chinese Academy of Sciences, Beijing 100049, China; 4School of Remote Sensing and Information Engineering, North China Institute of Aerospace Engineering, Langfang 065000, China

**Keywords:** the Belt and Road region, ecosystem quality, spatiotemporal changes, leaf area index, fractional vegetation cover, gross primary productivity

## Abstract

The Silk Road Economic Belt and the 21st Century Maritime Silk Road Initiative (BRI) proposed in 2013 by China has greatly accelerated the social and economic development of the countries along the Belt and Road (B&R) region. However, the international community has questioned its impact on the ecological environment and a comprehensive assessment of ecosystem quality changes is lacking. Therefore, this study proposes an objective and automatic method to assess ecosystem quality and analyzes the spatiotemporal changes in the B&R region. First, an ecosystem quality index (EQI) is established by integrating the vegetation status derived from three remote sensing ecological parameters including the leaf area index, fractional vegetation cover and gross primary productivity. Then, the EQI values are automatically categorized into five ecosystem quality levels including excellent, good, moderate, low and poor to illustrate their spatiotemporal changes from the years 2016 to 2020. The results indicate that the spatial distributions of the EQIs across the B&R region exhibited similar patterns in the years 2016 and 2020. The regions with excellent levels accounted for the lowest proportion of less than 12%, while regions with moderate, low and poor levels accounted for more than 68% of the study area. Moreover, based on the EQI pattern analysis between the years 2016 and 2020, the regions with no significant EQI change accounted for up to 99.33% and approximately 0.45% experienced a significantly decreased EQI. Therefore, this study indicates that the ecosystem quality of the B&R region was relatively poor and experienced no significant change in the five years after the implementation of the “Vision and Action to Promote the Joint Construction of the Silk Road Economic Belt and the 21st Century Maritime Silk Road”. This study can provide useful information for decision support on the future ecological environment management and sustainable development of the B&R region.

## 1. Introduction

The Belt and Road Initiative (BRI), which includes the “Silk Road Economic Belt” and the “21st Century Maritime Silk Road”, proposed by China in 2013, has become a great part of the practice of building a Human Destiny Community [1]. As one of the most influential international cooperation platforms, the BRI aims to promote the common development of regional trade and economy which assembles numerous developed and developing economies into an open and inclusive network of cooperation, providing unprecedented opportunities for international enterprises to explore new markets in countries along the routes [2,3]. Particularly, the implementation of the “Vision and Action to Promote the Joint Construction of the Silk Road Economic Belt and the 21st Century Maritime Silk Road (VAP)” by China in March 2015 has promoted mutually beneficial cooperation and achieved great progress [4,5]. However, for the unprecedented dimensions and the official major investment areas of the BRI, many scholars have shown concerns about its actual and potential negative influence on the ecological environment [6,7,8,9,10]. Therefore, ecological environment assessment over the Belt and Road (B&R) region has become crucial for sustainable development at both the regional and global scales.

In fact, the B&R region has a complex and fragile ecosystem, which mainly includes arid, semi-arid and sub-humid regions. Moreover, its response to intensified global climate change and human activity is predicted to become more sensitive [6,11]. Consequently, it is understandable that economic and social development may generate some inevitable influences on the ecological environment. Ecosystem quality is an important indicator for a sustainable ecological environment [12]. Therefore, objective and scientific ecosystem quality assessment is important for evaluating the impact of the BRI on the sustainable development of the B&R region and identifying the key areas for environmental protection, which would further maintain the health and stability of the ecological environment and promote regional sustainable development.

Although the scientific conceptualization of ecosystem quality has not formed a common consensus in the current studies [13], researchers believe that a perfect ecosystem is healthy and can meet the sustainable development of human beings’ needs, which is the important part of ecological environmental quality [14,15,16,17]. Vegetation is the most important component of the Earth’s ecosystem, which plays a key role in the circulation of energy, carbon and water, and its changes directly affect the climate, hydrology and soil of a region [18,19]. Therefore, vegetation is an important indicator of regional ecosystem quality, which can reflect the overall ecological environment change characteristics of a region. For this study, the definition of ecosystem quality as defined by the Ministry of Ecology and Environment of China was adopted. Ecosystem quality refers to the quality of natural vegetation in an ecosystem that uses vegetation indicators to diagnose whether the ecosystem is healthy under environmental stress and human disturbance within a specific time and space [20].

Remote sensing technology can objectively and quantitatively evaluate spatiotemporal changes in ecosystem quality, which has become a focus of ecosystem quality assessment at the regional scale. Many studies have conducted numerous ecosystem quality assessments based on remote sensing data, which have mainly focused on the assessment of ecological environmental quality, ecosystem function and ecosystem health using different indictors by integrating different kinds of products over different regions. For example, Wang et al. developed an eco-environmental quality index model using land use data, NDVI data from MODIS and statistical data to evaluate the ecological status of China and analyze its influencing factors during 2005–2010 [21]. Jing et al. constructed a remote sensing ecological index with principal component analysis for the rapid environmental quality assessment of the Ebinur Lake Wetland National Nature Reserve [22]. Fano et al. developed an eco-functional quality index using biotic data from three Italian coastal lagoons for the evaluation of environmental quality [23]. Zhang et al. proposed an ecosystem quality ratio by integrating the status of landscape structure, ecosystem services, ecosystem stability and human disturbance relative to their reference conditions and assessed the EQR changes in China’s counties from 1990 to 2015 [17]. However, previous studies have mainly focused on small-scale ecosystem quality assessment at the county, province and nation levels based on low-resolution remote sensing data. Ecosystem quality assessments and studies on spatiotemporal changes over the entire B&R area are still rare.

Furthermore, a technical specification for the investigation and assessment of ecosystem quality using vegetation parameters based on remote sensing data was released by the Ministry of Ecology and Environment of China on May 2021 [13]. However, the weighting of its indicators adopted average weighting which ignores the different importances of each indicator. Usually, multiple evaluation indicators are involved when evaluating the quality of regional ecosystems and the weights of each indicator largely influence the assessment result of the ecosystem’s quality [24]. Therefore, it is important to determine reasonable weights in assessing an ecosystem’s quality. Current weighting approaches include two groups: subjective and objective methods. Subjective methods require opinions from various experts to score indicators, such as analytic hierarchy process (AHP) methods, fuzzy methods, grey methods and artificial neural network methods. However, weights from these methods are directly affected by the subjective judgements of the experts. Objective weighting methods determine indicator weights based on the intrinsic structure of the data. Principle component analysis (PCA) is capable of projecting multivariate indicators into independent directions, which has been adopted in multiple ecosystem quality assessments and proven to be a robust technique in reducing the dimensions and extracting relationships among selected indicators [25]. Therefore, this study explores PCA methods to determine the weights of the indicators for an ecosystem quality assessment.

In order to make the ecosystem quality status and changes after the five years of the BRI, and especially the implementation of the VAP, explicit, this study proposes an objective and automatic remote sensing-derived method of ecosystem quality assessment and analyzes the spatiotemporal changes in the B&R region during 2016–2020. First, the ecosystem quality index (EQI) regarding vegetation status was obtained from three remote sensing ecological parameters including the leaf area index, fractional vegetation cover and gross primary productivity. Second, an assessment method was established to integrate the processes of objective indicator weighting, EQI calculating and categorizing. Third, the spatiotemporal ecosystem quality changes over the B&R region from 2016 to 2020 were analyzed and regions with significant changes were identified, which has seldomly been performed over the B&R region in the existing research.

## 2. Study Area

The B&R region was selected as the study area (Figure 1). Generally, the B&R region is an open area of international economic cooperation and is the longest spanning economic corridor in the world. To the east of the region is an active East Asian economic circle and to its west, a developed European economic circle, whereas the vast hinterland countries in the middle have great economic development potential. Presently, the B&R region includes 150 countries and 32 international organizations in Asia, Europe, Africa and Oceania as of December 2022. More and more countries will join the B&R region with the continuous expansion of national strategies [11,26]. In this study, the B&R region refers to the terrestrial part of the generalized B&R region, which contains all the ecosystem types in the terrestrial part of Eurasia, Africa and Oceania. In order to better analyze the ecosystem quality of the B&R region, the study area was divided into 10 geographical zones including West Asia, South Asia, East Asia, Southeast Asia, Central Asia, Russia, Australia, Europe, North Africa and Southern Africa, according to global ecological geographical zones and natural geographical locations [2,27].

The study area is vast and the ecosystem is complex with frequent changes in the ecological environment. The topography of the region is complex, including the world’s highest mountain in the Himalayas, as well as the Kunlun Mountains, Hindu Kush Mountains and other majestic mountains. It also includes the world’s highest Qinghai–Tibet Plateau, the Pamirs Plateau, the Iranian Plateau, the fertile plains and deltas of North China, the Ganges River and Eastern Europe. The climate differences are large, including tropical, temperate and cold zones which cross into a monsoon climate, continental climate and desert climate. It covers five climatic zones including a dry climatic zone, equatorial climatic zone, warm temperate climatic zone, cold temperate climatic zone and polar climatic zone [2,3]. It has rich vegetation and ecological resources owing to its large geographical span and differences in climate.

## 3. Methods

Figure 2 presents a flowchart of ecosystem quality assessments over the B&R region. Firstly, the indicators for ecosystem quality assessment were selected following scientificity and operability. Then, the relative density of each indicator was calculated and their weights were estimated using the principle component analysis (PCA) method, an objective weighting approach based on the intrinsic characteristics of data rather than subjective opinions from experts. Finally, an ecosystem quality index (EQI) was calculated and classified into five categorizes by the Jenks natural breaks method to provide a generalized understanding of the degree of ecosystem quality across the B&R region.

### 3.1. Indicator Selection and Data Processing

As an important component of ecosystems, changes in vegetation directly affect the climate, soil and other factors, which can well be reflected as change in an ecosystem’s quality [18]. Vegetation parameters estimated using remote sensing data have played a key role in environmental and ecological studies [19]. Therefore, it was reasonable to select vegetation parameters for evaluating the change in ecosystem quality. The leaf area index (LAI) is an essential parameter for characterizing the density of leaves and canopy structures, which can reflect the vegetation’s ability to undergo biophysical processes such as photosynthesis, respiration and transpiration and is an important input variable for carbon cycle models, crop growth models and water cycle models [28]. Fractional vegetation cover (FVC) is an important parameter for many weather prediction models, regional and global climate models, hydrological models and land surface models and has been extensively used in agriculture, soil erosion risk evaluation, drought monitoring and environmental assessment [29]. Gross primary productivity (GPP) denotes the production ability of vegetation’s photosynthesis [30]. In order to fully explore and utilize the information contained by vegetation, ecological parameters including the LAI, FVC and GPP were selected as the evaluation indicators in this study. In addition, land use and land cover (LULC) data were selected to distinguish between the vegetation ecosystem types.

Due to their better performance in spatiotemporal coverage and validation accuracy compared with other similar satellite products, LAI data were selected from China’s National Key R&D Program (Grant Number: 2020YFE0200700) and the FVC and GPP data were selected from Global Land Surface Satellite (GLASS) products. The LAI data, with a spatial resolution of 1 km and temporal resolution of quarterly, were generated from MODIS data and meteorological data using a BP neural network algorithm. The GLASS FVC and GPP data were obtained from the University of Maryland (http://www.glass.umd.edu/Download.html (accessed on 12 December 2022)) with a 500 m spatial resolution and 8-day temporal resolution in 2016 and 2020. The larger the LAI, FVC and GPP value, the better the vegetation growth state and the better the ecosystem quality. In addition, the LULC is also an important factor in ecosystem quality assessment [17], which reflects ecosystem types. The LULC data were also selected from China’s National Key R&D Program (Grant Number: 2020YFE0200700) with a spatial resolution of 1 km in 2016 and 2020, which were reclassified based on first-level land cover, including cultivated land, forest, grassland, shrub, wetland, tundra and non-vegetation types. The land cover data were generated using object-based multi-scale segmentation and change detection classification technology based on geographical region and sample migration. The overall validation accuracy by selecting points on high-resolution images was 88.97%, and the Kappa coefficient was 0.73 [27].

Ultimately, the ecosystem quality assessment was conducted at a 1 km spatial resolution and annual temporal resolution. So, the FVC and GPP indicators at 500 m of spatial resolution were resampled to 1 km using the nearest neighbor method. In addition, the LAIs with a quarterly temporal resolution and FVC with an 8-day temporal resolution were composited to annual datasets using the maximum value composite (MVC) method. 

### 3.2. Indicator Relative Density Calculation and Standardization

Indicator relative density (IRD) refers to the ratio of each indicator value to its reference value, which is a key step in calculating the EQI. A method joining eco-geographical zones with ecosystem types was used for the calculation of the IRD. In this study, according to the natural geographical environment, the B&R region was divided into 10 eco-geographical zones including the West Asia zone, South Asia zone, East Asia zone, Southeast Asia zone, Central Asia zone, Russia zone, Oceania zone, Europe zone, North Africa zone and Southern Africa zone. Otherwise, the vegetation ecosystem type contained the cultivated land ecosystem, forest ecosystem, grassland ecosystem, shrub ecosystem, wetland ecosystem and tundra ecosystem according to the LULC distribution in the B&R region.

Firstly, based on the eco-geographical zone and the ecosystem type of the B&R region, the indicators’ maximum values in each eco-geographical zone and ecosystem type were obtained as the reference value. The final IRD of each eco-geographical zone and ecosystem type for each year was calculated using Equation (1).
(1)$${IRD}_{i,j,k}=\frac{F_{i,j,k}}{F_{maxi,j,k}},$$
where $F_{i,j,k}$ and $F_{maxi,j,k}$ are the indicator value and its reference value for the *i*-th year, *j*-th eco-geographical zone and its *k*-th ecosystem type, respectively. 

Next, data standardization was conducted for all of the IRDs. The IRDs were calculated using different indicators that were distributed at different dimensions and dimensional units and would affect the data analysis. Therefore, the z-score method (Equation (2)) was used to normalize the IRDs to a uniform scale with mean values of 0 and a standard deviation of 1 which obeyed a standard normal distribution and is suitable for comprehensive comparative evaluation.
(2)$$Z=\frac{x-\bar{x}}{\sigma },$$
where Z  is the standardized IRD, *x* is the input original IRD, $\bar{x}$ is the arithmetic mean value of *x* and σ is the standard deviation of *x*.

### 3.3. Weighting Approach

Estimating the weights of all the standardized IRDs was another key step in the ecosystem quality assessment. The PCA method was used to estimate the weights of each standardized IRD in this study. PCA is one of the most commonly used unsupervised dimensionality reduction methods [25] and is a statistical analysis method that transforms multiple variables into a few principal components through dimensionality reduction technology. The weight estimation based on PCA is described as follows.

Firstly, suppose *X* is the input standardized *m* × *n* matrix with m observations and n variables. The data corresponding to each variable are recorded as *X*_1_, *X*_2_, *X*_3_…*X_n_*.
(3)$$X=\left[\begin{array}{ccc} x_{11} & \cdots & x_{1n}\\ \vdots & \ddots & \vdots \\ x_{m1} & \cdots & x_{mn} \end{array}\right]=\left[\begin{array}{ccc} X_{1} & \cdots & X_{n} \end{array}\right],$$

Then, the correlation coefficient matrix *R* is calculated by Equation (4) and the correlation coefficient is calculated by Equation (5).
(4)$$R=\left[\begin{array}{ccc} cov(X_{1},X_{1}) & \cdots & cov(X_{1},X_{n})\\ \vdots & \ddots & \vdots \\ cov(X_{n},X_{1}) & \cdots & cov(X_{n},X_{n}) \end{array}\right],$$
(5)$$r_{i,j}=cov(x_{ki},x_{kj})=\frac{\sum_{k=1}^{m}x_{ki}x_{kj}}{m-1},$$

The eigenvalues $\left(\begin{array}{c} \alpha_{1},\alpha_{2},\cdots ,\alpha_{n}, \end{array}\right)$ and the according eigenvectors $\beta =\left(\beta_{1},\beta_{2},\cdots ,\beta_{n}\right)\;$ are obtained by the Jacobian method and the contribution rate of the *i*-th PC $e_{i}$ is calculated by Equation (6). The greater the contribution rate, the stronger the information of the original variable contained in the PC.
(6)$$e_{i}=\frac{\alpha_{i}}{\sum_{i=1}^{m}\alpha_{i}},$$

Next, the top number *p* of PCs should be selected to satisfy the criterion $\sum_{i}^{p}e_{i}\geq \rho$ where ρ is determined based on research requirements (ρ = 0.85 in this study). Thus, the importance of each variable can be acquired by the top p columns in *C* and *e* as the following:(7)$$\gamma_{i}=\frac{{\sum_{i}^{m}\sum_{j}^{p}c_{i,j}e}_{j}}{\sum_{j}^{p}e_{j}},$$

The final weight $\omega_{i}\;$ for the *i*-th variable is calculated using Equation (8) and the sum of $\omega_{i}$ equal to 1.
(8)$$\omega_{i}=\frac{\gamma_{i}}{\sum_{i}^{p}\gamma_{i}},$$

All the standardized IRD weights were calculated for the years 2016 and 2020, and the final weights of each standardized IRD were obtained from the arithmetic means of the two years. The final weights of each standardized IRD were 0.3333, 0.2626 and 0.4041, respectively.

### 3.4. EQI Calculation and Classification

The final ecosystem quality index of each ecosystem type for each year was obtained by the weighted sum of the standardization IRDs.
(9)$$EQI=\sum_{i}^{m}\omega_{i}\times Z_{i},$$
where $\omega_{i}$ and $Z_{i}\;$ are the weight and standardized data for the *i*-th IRD, respectively, and *m* is the total number of IRDs in each eco-geographical zone and ecosystem type.

In order to achieve a basic generalization of the ecosystem quality over the study area and provide more intuitive knowledge for decision-making, the EQI values were graded into several categories showing the degree of the ecosystem’s quality. In this study, the Jenks natural breaks method [31] was used to classify continuous EQI values into five levels: excellent (EQI > 1.19), good (0.44 < EQI ≤ 1.19), moderate (−0.25 < EQI ≤ 0.44), low (−0.97 < EQI ≤ −0.25) and poor (EQI ≤ −0.97). The excellent level indicates an ecological environment with the highest vegetation coverage, the richest biodiversity and the most stable ecosystem, which is best suited for the survival of living things. The good level indicates an ecological environment with higher vegetation coverage and richer biodiversity. The moderate level indicates an ecological environment with moderate vegetation coverage, a general level of biodiversity and some uncomfortable restriction factors for living things. The low level indicates an ecological environment with poor vegetation coverage, severe drought, less rain and fewer species, which has obvious factors restricting the survival of living things. The poor level indicates ecological environment conditions that are harsh and a living environment that is harsh.

## 4. Results

### 4.1. Spatial Patterns of EQIs

The distributions of the continuous EQIs and EQI levels are presented in Figure 3 and Figure 4, respectively. EQI values varied from −1.9 to 2.9 in the year 2016 and from −1.9 to 2.8 in the year 2020, with the higher values indicating the better ecosystem quality of the study area throughout the study period. Generally, the spatial distributions of the EQI values were similar in the years 2016 and 2020, with irregular distributions. The lowest EQI values mainly appeared in Australia and south of Southern Africa. This may have been caused by the vegetation covers of those regions with sparse shrubland and grassland, which have low vitality and productivity. The lower EQI values were mainly distributed in the south of North Africa, north of West Asia, north of Central Asia, northwest of South Asia and northwest of East Asia. The reason may be that those areas are mainly located around deserts and wastelands with low vegetation coverage and low productivity. Moreover, low EQI values also appeared for the northeast of Russia, which may be a result of the low vegetation coverage with tundra located at low altitudes and severe climate conditions.

Figure 5 shows the proportions of EQI levels in the 10 eco-geographical zones of the B&R region in the years 2016 and 2020. In the study period, the proportions of the EQI levels in the 10 eco-geographical zones showed stable variations. The results showed that regions with the “low” level occupied the largest proportion with almost 30% whereas regions with “moderate”, “low”, and “poor” levels accounted for more than 70% of the entire study area. In addition, the regions with the proportions of “moderate”, “low” and “poor” levels exceeding 80% were distributed in Central Asia, North Africa, Australia and West Asia, while the regions with the proportions of “excellent” and “good” levels exceeding 80% were distributed in Southeast Asia only. This indicates that the ecosystem quality of the B&R region was relatively low and the EQI level across the B&R region can be defined as “low”. This also indicates that the regional development of ecosystem quality was unbalanced over the B&R region. Therefore, actions should be taken to improve the ecosystem quality of the B&R region, especially for the eco-geographical zones of Central Asia, North Africa, Australia and West Asia, which should pay more attention to the protection of the ecological environment for their more deteriorated ecosystem quality.

The proportions of each EQI level in the six ecosystem types were further analyzed for the years 2016 and 2020 (Figure 6). Over the study period, the proportions of the EQI levels in the six ecosystem types showed stable variation. The results showed that the regions with “excellent” and “good” levels exceeding 50% were distributed in forest ecosystems only, while the regions with “moderate”, “low” and “poor” levels exceeding 70% were distributed in the other five ecosystem types. This indicates that the ecosystem quality of the B&R region was relatively low and the ecosystem quality of the ecosystem type was unbalanced over the B&R region. Therefore, actions should be taken to improve the quality of the vegetation ecosystems for all the ecosystem types across the B&R region.

### 4.2. Temporal Changes in EQI

The differences in the continuous EQI values (symbolized by ΔEQI) from year 2016 to 2020 were calculated to identify the regions where the EQIs significantly changed according to the criterion of the “Regional ecological quality evaluation method” published by the Ministry of Ecology and Environment of the People’s Republic of China [32]. Finally, three significance levels were determined (Figure 7) according to the changed EQI value ranges: significantly increased EQI with ΔEQI > 0.05 (in green), no significant EQI changes with −0.05 < ΔEQI ≤ 0.05 (in yellow) and significantly decreased EQI with ΔEQI ≤ −0.05 (in red). 

Figure 8 shows the percentages of EQI changes in the 10 eco-geographical zones of the entire study area from 2016 to 2020. Combined with Figure 7, it can be concluded that regions with significantly decreased EQIs occupied 0.45% of the total area, mostly distributed north and central of Central Asia, southeast and central of Russia, southeast and southwest of Australia, and north of West Asia. Central Asia suffered the most serious deterioration in ecosystem quality, accounting for 0.15% of the total study area. This is in accordance with the review of ecological environment patterns across the B&R region performed by Zhang et al., which indicated that vegetation ecological factors will cause obvious zonal and gradient reduction on a scale of a few decades to a hundred years [11]. Generally, Central Asia has a temperate continental climate with scarce precipitation and widespread desert grassland, which makes the ecosystem quality of Central Asia poorer. However, regions with a significantly increased EQI occupied 0.22% and the regions with no significant EQI changes occupied 99.33% of the entire study area, which indicates that most of the B&R regions showed no significant change from the years 2016 to 2020.

Figure 9 shows that the percentages of each ecosystem type area with different EQIs changed from 2016 to 2020 in the four eco-geographical zones, which had significantly decreased EQIs. Generally, the total significantly decreased EQI areas in certain eco-geographical zones are the sum of the significantly decreased EQI areas in each ecosystem type. In the four eco-geographical zones, the cultivated land and grassland experienced significantly decreased EQIs. The forest ecosystem type suffering from significantly decreased EQIs was distributed in Russia and Australia. The shrubland ecosystem had a significantly decreased EQI distributed in Russia. Overall, the cultivated land in Central Asia suffered the most serious deterioration in ecosystem quality in the study period. This is in accordance with the articulate ecosystem in Central Asia that was found to be aggravated by Jiang et al. 2022, who pointed out that the interplay between anthropogenic forces and the natural variability in the Interdecadal Pacific Oscillation exacerbated the agricultural drought in Central Asia for nearly 30 years [33].

## 5. Discussion

### 5.1. The Spatiotemporal Changes in EQI in the B&R Region

The overall ecosystem quality level of the B&R region can be considered as “low”, since this designation accounts for the largest area proportions for almost 30% ofthe study period. The spatial distributions of EQI in the B&R region are in accordance with the distribution of vegetation ecosystem types and non-vegetation areas, especially the desert ecosystem, and they present an irregular pattern for the entire B&R region. The areas with the lowest EQIs are mainly distributed in Australia, south of Southern Africa, the junction between Southern Africa and North Africa, north of West Asia, north of Central Asia, and northwest of East Asia, which indicates that the ecosystem quality of those areas is poor. These areas are consistent with the ecological restoration project implemented in regions or countries summarized by the FAO [34]. They are mainly located around arid and semi-arid areas, which belong to a semi-desert climate with little precipitation throughout the year [35]. The annual precipitation in these areas is generally less than 250 mm. In addition, the large daily temperature range, accompanied by frequent sand activity, can lead to a bare and sparse vegetation ecosystem with the lowest coverage and productivity. Northeast Russia has the lowest EQI and is located in the Arctic Circle, belonging to the cold tundra climate. The altitude and climate conditions result in scarce precipitation and weak solar radiation. The vegetation ecosystem is mainly sparse tundra with lower coverage and productivity [36]. The zones including Southeast Asia, east of East Asia, central of Southern Africa, southwest of Russia and south of Europe have the highest EQIs. The regions south of Southeast Asia and central of Southern Africa have a tropical rainforest climate, and those north of Southeast Asia and east of East Asia belong to a tropical monsoon climate. These climatic conditions make the areas suitable for vegetation growth and production, leading to an excellent ecosystem quality. Southwest Russia is located in a cold temperate zone with insufficient heat, but the snowfall is very rich. Therefore, a large area of cold evergreen coniferous forest has been developed, leading to a rich annual cumulative productivity. As for south of Europe, the warm and humid air from the Atlantic can penetrate into the inland, bringing in a warm climate and relatively high precipitation of more than 1000 mm. Moreover, the climate conditions boost vegetation growth and production, leading to a better ecological environment.

With regard to the temporal patterns established by the change analysis, all 10 eco-geographical zones experienced no significant EQI changes with a proportion of over 90% remaining stable. The regions with a significantly decreased EQI, which accounted for 0.45%, are mainly distributed in grassland and farmland ecosystems in arid and semi-arid areas. These areas are more vulnerable to increasing climate change and human activities than other terrestrial vegetation, which will experience varying degrees of degradation in the context of global climate change [37,38,39]. Therefore, it can be deduced that climate change is the main reason for the degradation of vegetation ecosystems and there is no obvious relativity between the conduction of the BRI and the ecosystem quality changes from 2016 to 2020. Subtle impacts still need to be investigated with more data including but not limited to climate data and sociopolitical data for small-area analysis in future work.

### 5.2. Advantages and Limitations of the Proposed Method

This study proposes an objective and automatic method for ecosystem quality assessment and analyzed the spatial and temporal distribution characteristics of ecosystem quality over the B&R region at a 1 km spatial resolution scale. The method is effective in integrating the vegetation growth status derived from vegetation ecological parameters for large regional ecosystem quality assessment. One advantage of the proposed method is that the indicators selected are simple and easy to obtain and can quickly assess ecosystem quality in large regions. Another advantage is that it introduced using the PCA approach to determine the most reasonable weighting of the indicators in generating ecosystem quality results with higher objectivity and credibility. The method is different from previous studies that evaluate ecosystem quality using average weighting methods based on expert opinions [40]. However, the vegetation ecological parameters of LAI, FVC and GPP represent different characteristics of vegetation status and have different influences on ecosystem quality. Therefore, it is necessary to determine which indicator occupies a larger proportion in the evaluation of ecosystem quality. The introduced PCA weighting approach identified that the GPP was more important for determining ecosystem quality, which was reasonable because the GPP is directly related to the ability of vegetation to photosynthesize. Furthermore, the PCA weighting approach also accelerates the weighting process without requiring experts’ consultation. Therefore, this improved method of ecosystem quality assessment was better in this study at dealing with multiple indicators and effectively estimating indicator weights based on the intrinsic features of the indicators compared with subjective methods, generating a reasonable result of ecosystem quality assessment in the B&R region. In addition, this study analyzed the spatiotemporal patterns of ecosystem quality over the entire B&R region and identified the changes from the years 2016 to 2020, which has not been provided by previous research. The results can objectively reflect the spatial distribution and temporal changes in the ecosystem quality of the study area. They can also provide scientific information services and give a credible reference for the sustainable development of the B&R region’s regional ecology.

However, there were also some limitations to this study, even though it achieved a reliable assessment of ecosystem quality. Firstly, this study selected vegetation ecological parameters as the indicators for assessing ecosystem quality, while not taking into consideration other factors such as human stress, social and political conditions, which could be considered in future work. Secondly, due to the limitations of data availability, this study did not consider more aspects related to natural hazards and public health emergencies like the COVID-19 pandemic, which will also be taken into consideration for future studies. Finally, the B&R region is a relatively large region for ecosystem quality assessment in that it is difficult to fully identify the ecosystem processes at play with 1 km resolution data. Therefore, future studies will be conducted over the regions with a significantly decreased EQI to further explore the causes of the ecosystem quality changes based on time series and higher spatial resolution remote sensing data. 

## 6. Conclusions

This study developed an objective and automatic method for evaluating ecosystem quality over the entire B&R region based on ecological parameters derived from remote sensing data. This method solved the problem of the difficulty in quantitatively evaluating the quality of large regional ecosystems and achieved a reliable ecosystem quality assessment, which is portable and suitable for ecosystem quality assessments in any other regions. In addition, this study analyzed the spatial distribution and temporal change patterns of EQI over the B&R region from the years 2016 to 2020. The analysis indicated that the ecosystem quality over the B&R region was relatively low and had no significant changes after the conduction of the BRI, especially including the implementation of the VAP. This study can provide useful information for future ecological environment management projects and the sustainable development of the B&R region.

## Figures and Tables

**Figure 1 sensors-23-07752-f001:**
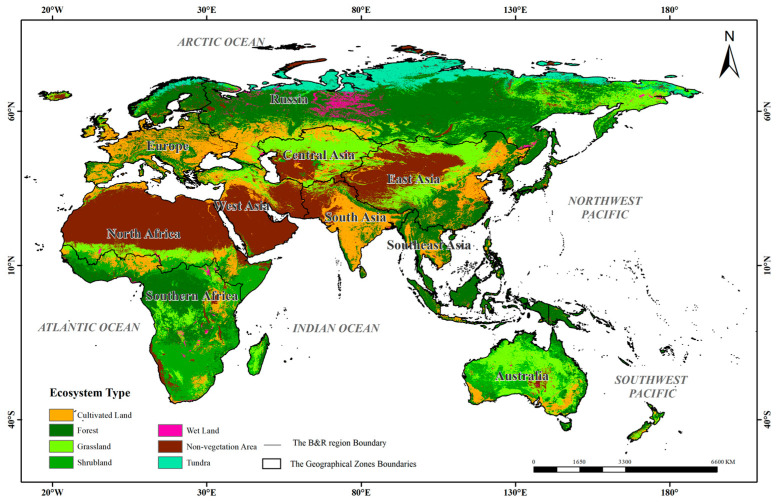
The geographical location of the study area.

**Figure 2 sensors-23-07752-f002:**
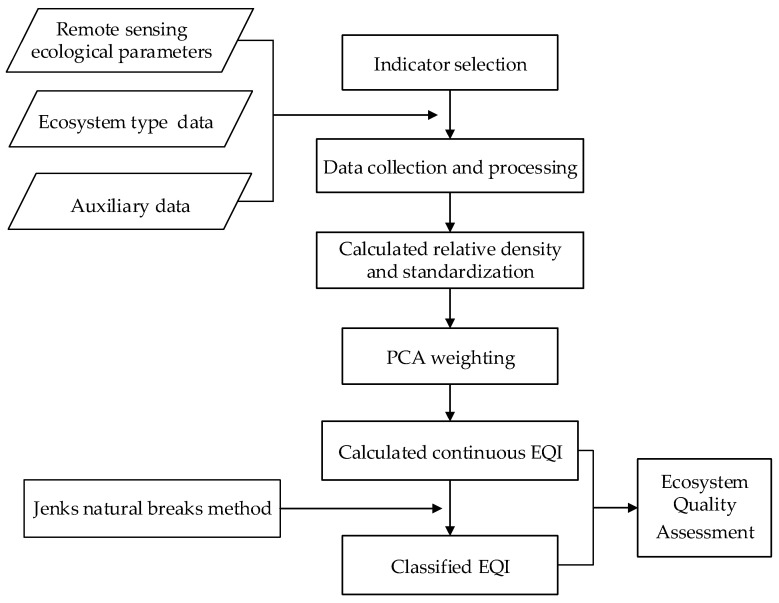
Flowchart of the ecosystem quality assessment over the B&R region.

**Figure 3 sensors-23-07752-f003:**
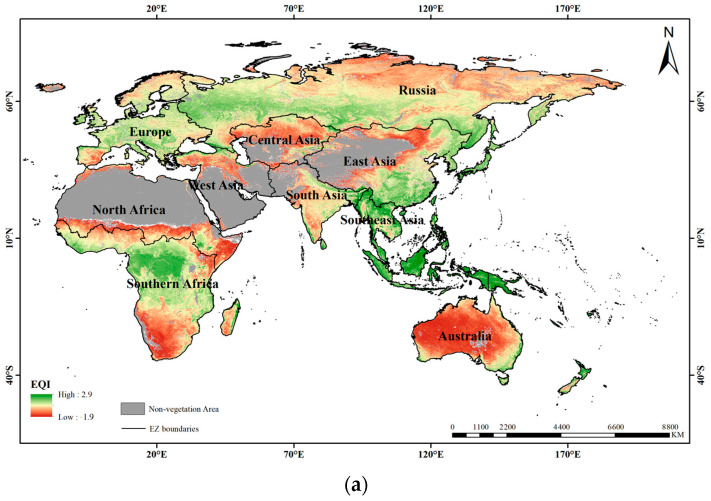
Continuous EQI distributions in (**a**) 2016 and (**b**) 2020.

**Figure 4 sensors-23-07752-f004:**
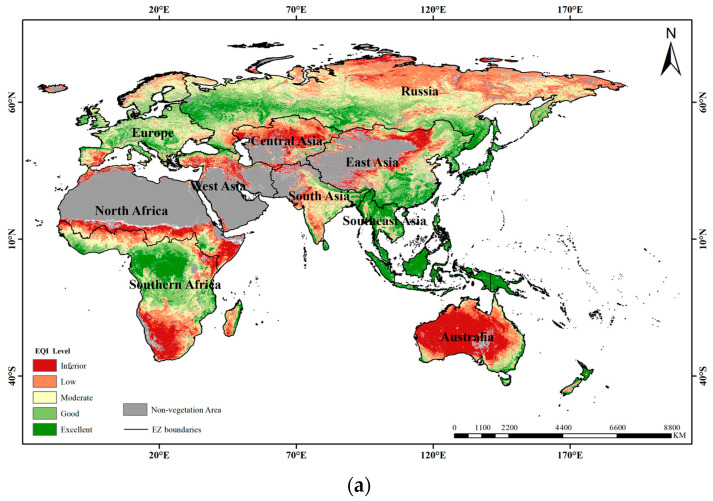
EQI level distributions in (**a**) 2016 and (**b**) 2020.

**Figure 5 sensors-23-07752-f005:**
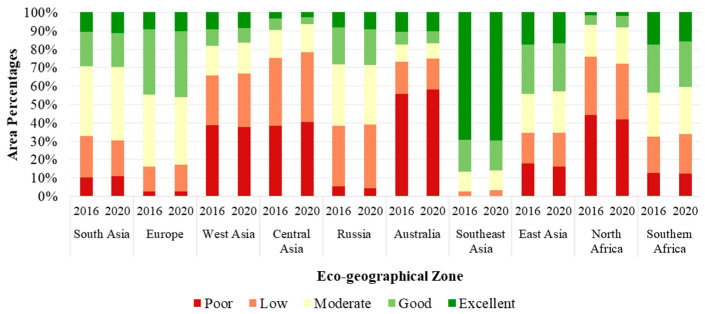
Proportions of EQI levels in each eco-geographical zone in 2016 and 2020.

**Figure 6 sensors-23-07752-f006:**
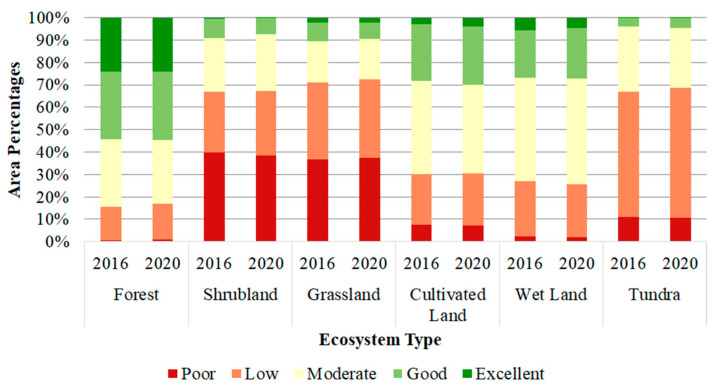
Percentages of EQI levels in the six ecosystem types in 2016 and 2020.

**Figure 7 sensors-23-07752-f007:**
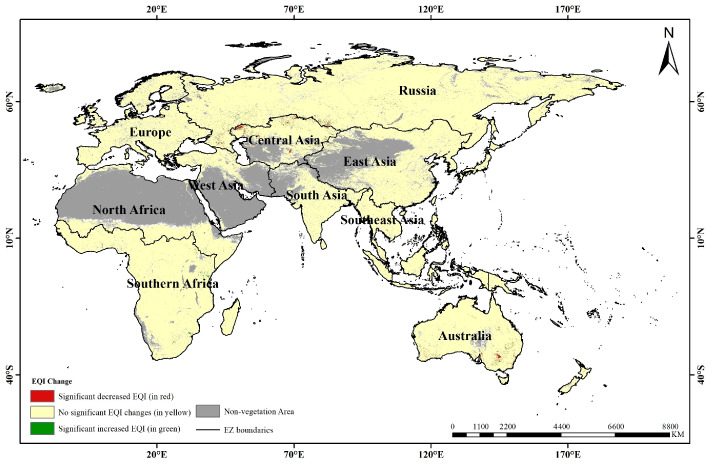
Regions with no significant EQI changes (in yellow), significantly increased EQI (in green) and significantly decreased EQI (in red) from 2016 to 2020.

**Figure 8 sensors-23-07752-f008:**
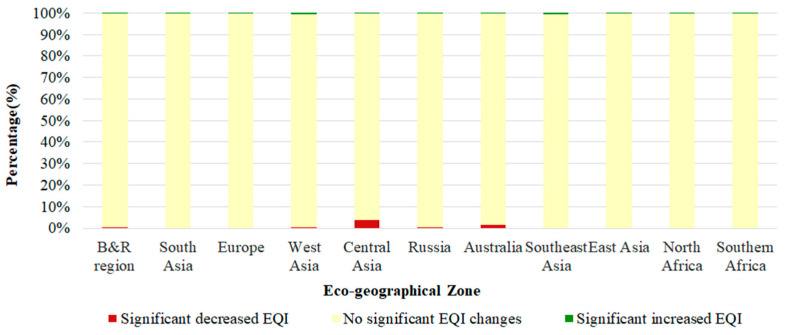
Percentages of EQI changes in the 10 eco-geographical zones of the entire study area from 2016 to 2020.

**Figure 9 sensors-23-07752-f009:**
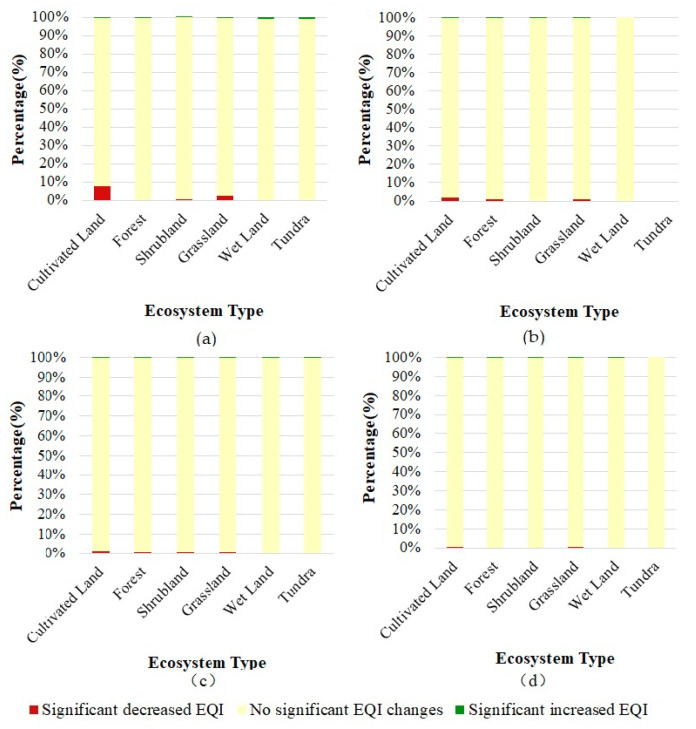
Percentages of each ecosystem type area with different EQI changes from 2016 to 2020 in the four eco-geographical zones with a significantly decreased EQI: (**a**) Central Asia, (**b**) Australia, (**c**) Russia, (**d**) West Asia.

## Data Availability

Not applicable.

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
