# Peer review of "Spatio-Temporal Changes in Ecosystem Quality across the Belt and Road Region"

_sensors, 2023, doi:10.3390/s23187752_

Round 1

Reviewer 1 Report (Previous Reviewer 1)

Revised Title: Assessment of Ecosystem Quality Changes in the Belt and Road Region Over Time

I previously revised the manuscript and faced significant challenges, particularly related to the methodology. This resubmission features a restructured format, primarily focusing on changes within the discussion and conclusions sections, while the core components remain unchanged. As a result, many of the observations and suggestions I previously made were not incorporated:

The authors propose an objective and automated approach for evaluating ecosystem quality and examining spatio-temporal variations across the Belt and Road (B&R) region.

The research, centered on the analysis of Ecosystem Quality Index (EQI) patterns from 2016 to 2020, revealed that nearly 99.33% of regions displayed insignificant EQI alterations.

The authors assert that these findings offer valuable insights for guiding decisions concerning future ecological management and the sustainable progress of the B&R region.

However, persisting issues with the manuscript remain. Primarily, the study's geographic scope is inadequately defined on the maps and within the text.

Questions arise regarding the precise dimensions of the Belt and Road (B&R) region and why this spatial extent isn't delineated on the maps.

This central issue persists: The study's reliance on 1 km by 1 km pixels could contribute to the minimal disparities observed between 2016 and 2020, as subtle changes might not be discernible at this resolution. Consequently, this methodology's efficacy in detecting minor changes is questionable. Additionally, there's a conspicuous absence of calibration or error analysis. For validation purposes, it's crucial to scrutinize if known impacts are accurately identified using this analytical approach.

All figures remain unchanged, indicating an absence of responses to my prior observations.

None of the maps elucidate the nuanced differences between the two years at a finer scale.

Figure 1: Geographic Localization of the Study Area.

It's imperative to depict the geographic boundaries of the study area on the maps.

A comprehensive analysis should consider variations across distinct geographic zones and climates, rather than applying generalizations to the entire region. Socio-political dynamics may differ significantly and should be linked to observed changes.

To yield meaningful insights, the results should be dissected into smaller units. For instance, Russia and Australia encompass vastly diverse geographic and climatic conditions.

The conclusions are overly concise, essentially echoing the results. Enhancements are warranted to provide a more comprehensive summary.

In conclusion, these revisions aim to address the concerns I previously raised while providing a more polished version of the manuscript.

Author Response

We highly appreciate you for your valuable comments that helped us improve our manuscript. We have provided a point-to-point response (in red font) to all comments below. We hope that the revised manuscript can obtain your acceptance.

Reviewer 2 Report (New Reviewer)

The authors propose an objective and automatic method to assess the ecosystem quality and analyze the spatio-temporal changes of the B&R region. This is an interesting and rigorous paper. This study provides a new idea for the assessment of ecosystem quality. The results can provide valuable reference for the ecological environment management and the sustainable development of the B&R region. Overall, I felt the paper was well structured and quite easy to follow. The results seem to be quite good. The English is generally quite good and it was possible for me to understand everything in the paper. But I do have some minor revisions, which are listed below.

 1)     Fig.4 shows the EQI level distributions in 2016 (a) and 2020 (b). But your legend is marked with EQI. I suggest amending it to EQI level.

 2)     P9, L307, “…shows the proportions of EQI category…”, Fig.5 and Fig.6 “Proportions of EQI category…” ,I can understand that it refers to the EQI level. I think there are some difference between “category” and “level”.  In fact, the EQI level is more suitable for describing ecosystem quality grade. I suggest revise it.

3)     It is precise that the results were discussed with respect to different geographic areas and climates of each area. It can be understand that natural hazards maybe have a certain impact on ecosystem quality. How the public health emergencies like COVID-19 pandemic effect the EQ? Could you explain it?

Author Response

We highly appreciate you for your valuable comments that helped us improve our manuscript. We have provided a point-to-point response (in red font) to all comments below. We hope that the revised manuscript can obtain your acceptance.

Reviewer 3 Report (New Reviewer)

The work is interesting and useful. The manuscript is well organized.

Here are a few suggestions for improvement of manuscript.

1. The definition of ecosystem quality in line 71 does not cite literature.

2. It is suggested to add why to choose ecosystem quality from 2016 to 2020 as the research object in the introduction.

3. To ensure the observability and richness of the picture, it is recommended to include iconic sites and climatic zones in Figure 1.

4. Based on the seven ecosystem types in the Belt and Road region in Figure 1, it is recommended to include Non-vegetation Area in line 219.

5. It is recommended to unify the formatting of formulas in articles.

6. It is proposed to cancel the bolding of text in line 325.

7. Figure 8 does not reflect the specific distribution of the four Regions with significant decreased EQI in line 346 , and it is recommended to modify it.

8. It is recommended to include the text "from 2016 to 2020" in the title of Figures 8 and 9.

9. It is recommended that the data be refined and data of little relevance to the study be removed.

10. Please strengthen the explanation of the conclusion section. It is recommended that quantitative reasoning be used to compare with appropriate benchmarks.

11. The Discussion section needs to add available reasoning to the discussion of the results. It is suggested that the authors add some reasoning and comparisons based on the combined literature.

Author Response

We highly appreciate you for your valuable comments that helped us improve our manuscript. We have provided a point-to-point response (in red font) to all comments below. We hope that the revised manuscript can obtain your acceptance.

Round 2

Reviewer 1 Report (Previous Reviewer 1)

This manuscript should include a supporting file with all relevant information in tables because changes between 2016 and 2020 a difficult to identify in the figures. 

Minor editing of the English language is required, for example:

line  186:  Global LAnd Surface Satellite (GLASS)

should be Global Land Surface Satellite (GLASS)

line: 189    University of Mar-  yland    should be 

University of Mary-  land  

line 289:   The reason maybe that those areas mainly located around de- serts and wastelands with low vegetation coverage and low productivity.

should be:  The reason may be that those areas are mainly located around de- serts and wastelands with low vegetation coverage and low productivity

Author Response

Thank you for your valuable comments. We have revised manuscript and provided a point-to-point response (in red font) to all comments below.

Reviewer 3 Report (New Reviewer)

The quality of manuscript at present is well.

Author Response

Thank you again for your valuable comments.

This manuscript is a resubmission of an earlier submission. The following is a list of the peer review reports and author responses from that submission.

Round 1

Reviewer 1 Report

Review of Spatio-temporal Changes of ecosystem quality across the Belt and Road Region                                                                                  
The authors state that the study proposes an objective and automatic method to assess the ecosystem quality and analyzes the spatio-temporal changes of the B&R region.

The study, which is based on the EQI pattern analysis between the years 2016 and 2020, interpreted that the regions with no significant EQI change accounted for up to 99.33% and most regions with significantly decreased EQI were not located along the construction line of the main projects promoted by BRI.

The authors claim that the results are useful information for decision support on the management of the future ecological environment and the sustainable development of the B&R region.                                                             31

There are some principal problems with the study. First, the study area is not well defined in the maps and is not well described.

 What are the exact dimensions of the Belt and Road (B&R) region, why is the area not indicated in the maps?

Line 87, 88

You state that previous studies mainly focus on small-scale EQ assessment for county, province, and nation. The EQ assessments and their spatio-temporal changes over the entire B&R area are still rare.

The study is based on 1km by 1 km pixels, which seems to be one of the reasons for the little difference observed between 2016 and 2020 because smaller structures may not be recognized. So it seems that this methodology is not adequate to recognize minor changes. There is no calibration or error analysis included. For example, some known impacts should be analyzed to verify if they can be recognized adequately as well by this analysis.

No map indicates the differences between both years on a more detailed scale. The observed changes should be analyzed with respect to other factors like geographical, political,or climatic aspects.

Line 150 151

Figure 1: The geographic location of the study area.

The geographic location of the study area must be indicated in the maps.  

The results should be discussed with respect to different geographic areas and climates, and not for the whole area, and the different social and political conditions may vary and should be discussed with respect to the observed changes.

Figure 5. Proportions of EQI category in each eco-geographical zone in 2016 and 2020        

The results should be analyzed with respect to smaller units, for example, Russia and Australia are far too big and include very different geographic areas and climates.

The conclusions are too short and only repeat the results, they must be improved.

Moderate editing of the English language is required, a native English speaker should correct the text.

Reviewer 2 Report

This manuscript entitled "Spatio-temporal changes of ecosystem quality across the Belt and Road region" provides useful information to support decision-making on future ecological management and sustainable development in the B&R area. The authors present the objective evaluation of the research and the direction of development based on it. It is considered that the overall content is well-organized.

Reviewer 3 Report

Dear Authors,

In l. 109-111 the aim of the study was outlined. The Authors declare “… this study aims to propose an objective and automatic remote sensing derived method of ecosystem quality assessment and analyze the spatio-temporal changes of the B&R region during 2016-2020“. A Reader can expect at least:
1. For validation, in selected regions a comparison of the results from automatic remote sensing with the results from classical methods of the ecosystem quality assessment should be presented.
2. Application of the formulated data elaboration method for ecosystem quality assessment in credibly defined B&R region should be presented.
3. Application of the formulated data elaboration method for ecosystem quality assessment in appropriately defined period should be presented.
In the article the Reader finds (or not):
Re 1. No actual information about comparison of the results from the new and the traditional methods are presented,
Re 2. For B&R region 3 continents were selected. It is clearly an overestimation, impact of traffic on the roads and activity of the associated infrastructure isn’t so spatially extended.
Re 3. Selection of 2020 for comparison with 2016 is unfortunate. The effects of COVID-19 pandemic include retardation of industrial and social activity. As a result significant reduction in production of atmospheric pollutants was observed. The overall quality of the natural environment became higher.
Some other drawbacks of the article were noticed:
1. Poor discussion about possible influence of different factors on EQI.
2. As a result of low number (2) of time sections, rare or incidental specific features affecting EQI could be regarded as typical.
3. No discussion about the parameters’ uncertainty was presented.

Finally, the conclusion “… it could conclude that the BRI and the implementation of “Vision and Action to Promote the Joint Construction of the Silk Road Economic Belt and the 21st Century Maritime Silk Road” did not cause negative impact to the ecosystem quality of the B&R region in the study period” isn’t supported by the presented data and results of their analysis.